# Enhanced Photovoltaic Performance in D-π-A Copolymers Containing Triisopropylsilylethynyl-Substituted Dithienobenzodithiophene by Modulating the Electron-Deficient Units

**DOI:** 10.3390/polym11010012

**Published:** 2018-12-21

**Authors:** Junfeng Tong, Lili An, Jie Lv, Pengzhi Guo, Xunchang Wang, Chunyan Yang, Yangjun Xia

**Affiliations:** 1School of Materials Science and Engineering, Lanzhou Jiaotong University, Lanzhou 730070, China; zoeyejie@163.com (J.L.); YChY-5@126.com (C.Y.); 2School of Chemical Engineering, Northwest Minzu University, Lanzhou 730030, China; anlili2011@163.com; 3National Green Coating Technology and Equipment Research Center, Lanzhou Jiaotong University, Lanzhou 730070, China; shxygpz@126.com; 4CAS Key Laboratory of Bio-Based Materials, Qingdao Institute of Bioenergy and Bioprocess Technology, Chinese Academy of Sciences, Qingdao 266101, China; wang_xc@qibebt.ac.cn

**Keywords:** TIPS-substituted dithienobenzodithiophene, modulating the electron-deficient units, naphtho[1,2-*c*:5,6-*c*′]bis[1,2,5]thiadiazole, fluorination, photovoltaic property

## Abstract

Three alternated D-π-A type 5,10-bis(triisopropylsilylethynyl)dithieno[2,3-*d*:2′,3′-*d*′]-benzo[1,2-*b*:4,5-*b*′]dithiophene (DTBDT-TIPS)-based semiconducting conjugated copolymers (CPs), PDTBDT-TIPS-DTBT-OD, PDTBDT-TIPS-DTFBT-OD, and PDTBDT-TIPS-DTNT-OD, bearing different A units, including benzothiadiazole (BT), 5,6-difluorinated-BT (FBT) and naphtho[1,2-*c*:5,6-*c*′]-bis[1,2,5]thiadiazole (NT), were designed and synthesized to investigate the impact of the variation in electron-deficient units on the properties of these photovoltaic polymers. It was exhibited that the down-shifted highest occupied molecular orbital energy level (*E*_HOMO_), the enhanced aggregation in both the chlorobenzene solution and the solid film, as well as the better molecular planarity, were achieved using methods involving fluorination and the replacement of BT with NT on the polymer backbone. The absorption profile was little changed upon fluorination; however, it was greatly broadened during replacement of BT with NT. Consequently, the optimized photovoltaic device based on the PDTBDT-TIPS-DTNT-OD exhibited synchronous enhancements in the open-circuit voltage (*V*_OC_) of 0.88 V, the short-circuit current density (*J*_SC_) of 7.21 mA cm^−2^, and the fill factor (*FF*) of 52.99%, resulting in a drastic elevation in the PCE by 129% to 3.37% compared to that of the PDTBDT-TIPS-DTBT-OD. This was triggered by PDTBDT-TIPS-DTNT-OD’s broadened absorption, deepened *E*_HOMO_, improved coplanarity, and enhanced SCLC mobility (which increased 3.9 times), as well as a favorable morphology of the active layer. Unfortunately, the corresponding PCE deteriorated after incorporating fluorine into the BT, due to the oversized aggregation and large phase separation morphology in the blend films, severely impairing its *J*_SC_. Our preliminary results demonstrated that the replacement of BT with NT in a D-π-A type polymer backbone was an effective strategy of tuning the molecular structure to achieve highly efficient polymer solar cells (PSCs).

## 1. Introduction

Solar energy has gained considerable attention because it is considered environmentally friendly, is inexhaustible, and has a widespread distribution [1,2]. State-of-the-art bulk heterojunction (BHJ) polymer solar cells (PSCs), composed of conjugated polymers (CPs) as donors and fullerene derivatives as acceptors, can directly convert sunlight into electricity energy. This has led to a focus on PSCs because they possess unique merits, including low-cost fabrication of large-area devices, light weight, mechanical flexibility, and easy tunability of the PSCs chemical properties [3,4,5]. Recently, increased efforts have been devoted towards boosting the development of PSCs, where power conversion efficiencies (PCEs) in the single-junction have reached up to 11–13% [5,6,7,8]. However, in terms of commercialization, further improvement to the PCE is still needed, which can be achieved by developing efficient polymer donors for the BHJ photoactive layers. One of the most efficient methods for elevating the PCE is the development of low band gap (LBG) CPs, which can harvest more sunlight to generate a large short-circuit current (*J*_SC_), theoretically, in the visible and near infrared (IR) regions [4,9,10,11]. Meanwhile, a deeper highest occupied molecular orbital (HOMO) energy level (*E*_HOMO_) of donor CPs and a suitable lowest unoccupied molecular orbital (LUMO) energy level (*E*_LUMO_) of the fullerene acceptor, can ensure a high open-circuit voltage (*V*_OC_) and efficient exciton dissociation at the polymer/fullerene interface, respectively [12,13]. Moreover, constructing a planar molecular skeleton is of great importance and significance to obtain a closer π–π stacking, which can adjust the internal aggregation, charge transfer ability, and molecular conjugation [2]. Furthermore, it is essential to have appropriate compatibility with the fullerene acceptor to form a nanoscale bicontinuous interpenetration network to facilitate efficient charge generation and transport [14,15]. To address these goals, it is well established that the most popular and successful strategy is the incorporation of electron rich moieties (D) and electron deficient moieties (A), and alternating them onto the backbone. This can effectively influence the optical property and band gap, electronic energy levels, charge mobility, crystallinity, and morphology to promote the *V*_OC_, *J*_SC_, and the fill factor (*FF*) of the devices, and hence the PCE by selecting suitable D and A units and fine modulating the intramolecular charge transfer (ICT) interaction [16,17]. Nevertheless, the judicious selection of appropriate D and A units for constructing efficient donor CPs is still a great challenge.

Amongst the numerous A building blocks reported, benzo[*c*][2,1,3]thiadiazole (BT) containing a 1,2,5-thiadiazole ring and a strong *o*-benzoquinoidal unit has been extensively focused on owing to its easy synthesis, outstanding electro-withdrawing ability, and compact planar structure, which are conducive to the electron delocalization in the CPs after BT is incorporated into the polymer backbone [18,19,20,21]. With these good features, excellent D–A photovoltaic (PV) CPs containing BT, such as PCDTBT [22], PCPDTBT [23], PBTz4T [5,14,24], PffBT4T-2OD [5], PBDT-DTBT [25], and PDTBDT-BT [26], have been widely investigated. Moreover, incorporating fluorine (F) onto the BT has also proven to be an effective and facile strategy when targeting efficient PSCs, which can effectively down-shift the energy level without sacrificing the bandgap owing to the electronegative element of the F atom with a Pauling electronegativity of 4.0, reducing the undesirable steric hindrance due to the relatively small van der Walls radius (147 pm), as well as promoting the fascinating molecular ordering assisted by the inter- and intra-molecular C–F···H, F···S, and C–F···π_F_ interaction. Therefore, this accelerates the exciton dissociation and increases the lifetime of the charge carrier by reducing the Coulombic interaction between the electron–hole by virtue of the induced dipole [27,28,29,30,31,32,33,34,35,36]. The fluorination had different effects on the absorption properties in different CPs systems, including blue-shift [37,38,39,40], without change [41,42,43,44], and red-shift [45,46]. What cannot be ignored is that fluorination can improve the coplanarity of the molecular backbone in CPs, and it further improves the charge mobility; however, it inevitably reduces the solubility of CPs to restrict the solution-processed fabrication [40]. As an alternative, an emerging centrosymmetric naphtho-[1,2-*c*:5,6-*c*′]bis[1,2,5]thiadiazole (NT), with an extended conjugation, an enlarged planarity, and a stronger electron-withdrawing ability due to the existence of doubly fused 1,2,5-thiadiazole rings relative to the axisymmetric BT, exhibited interesting electronic properties and a high self-assembling nature [7,21,24,25,40,47,48,49,50]. Early in 2011, Huang et al. first incorporated NT into the polymer backbone and combined it with alkylthienyl-substituted benzo[1,2-*b*:4,5-*b*′]dithiophene (BDT) to develop PBDT-DTNT, showing an obviously extended absorption profile, which had a one order of magnitude higher charge mobility and a 2.84 times higher PCE compared to its BT-based counterpart PBDT-DTBT, even if the decreased *V*_OC_ (0.8 V *vs*. 1.0 V) originated from an elevated *E*_HOMO_ [25]. After that, Osaka et al. combined NT with oligothiophene, and the designed PNTz4T exhibited a narrower bandgap, a deeper *E*_HOMO_, a more highly ordered structure, and a 2.48 times increase in the PCE, compared to those of its counterpart PBTz4T [24]. Xu et al. further introduced 4,9-di(thien-2-yl)naphthobisthiadiazole (DTNT) and 4,7-dithienyl-benzothiadiazole (DTBT) as a 2D conjugated side chain into the BDT-based polymer backbone, and found that PBDTT-TANT could induce a stronger ICT effect between the polymer backbone and the NT-based conjugated side chain. Additionally, an overtly broadened absorption spectra and a tighter molecular stacking structure were achieved, and thus the *J*_SC_, *FF*, and PCE correspondingly improved from 11.12 to 13.06 mA cm^−2^, 62.1% to 65.2%, and 6.74% to 8.04%, respectively, despite a lower *V*_OC_ value [50]. These results confirmed that the replacement of BT with NT was a facile and effective strategy and it was expected to broaden the absorption, tune the energy level, and optimize the film morphology, thereby promoting the PV performance.

Recently, large heteroacenes have been employed in efficient PSCs, as they provide several advantages, including improved charge mobility originating from the increased interchain π–π stacking assisted by large planar π-conjugated structures [51]. In addition, an increased effective conjugation length in heteroacenes-containing CPs is expected to minimize the Marcus reorganization energy of the polymer system to facilitate exciton separation into free charge carriers [51,52,53,54,55,56,57,58]. Specifically, dithieno[2,3-*d*:2′,3′-*d*′]benzo[1,2-*b*:4,5-*b*′]dithiophene (DTBDT) has proven to be an excellent building block in the PSCs field [59,60,61,62,63]. Furthermore, bulk triisopropyl-silylethynyl (TIPS) on heteroacenes not only improve the solubility, crystallinity, and oxidative stability of semiconductors, but they also promote the π-orbital overlap between conjugated molecules [17,54,59,60,61,62,63]. In 2013, Kim and a co-worker introduced TIPS into BDT, where they prepared two benzo[*d*][1,2,3]triazole-based copolymers, and demonstrated that TIPS could not only effectively deepen the *E*_HOMO_ from −5.26 to −5.40 eV and correspondingly elevate the *V*_OC_ from 0.77 to 0.80 V, but they could also impel formation of a crystalline lamellar structure to enhance the hole mobility and increase the *J*_SC_ (12.69 vs. 7.87 mA cm^−2^), *FF* (55% vs. 48%), and PCE (5.53% vs. 2.88%) of the devices [61]. Afterwards, our group designed a different side chain alkyloxy (OR), alkylthienyl, and TIPS onto a DTBDT moiety, and we also found that a DTBDT-TIPS-based CP yielded the best PCE of 6.39% via simultaneous enhancements in the *V*_OC_, *J*_SC_ and *FF* [54].

Prompted by the abovementioned considerations, this study focused on three D-π-A type CPs, PDTBDT-TIPS-DTBT-OD, PDTBDT-TIPS-DTFBT-OD, and PDTBDT-TIPS-FTNT-OD (Figure 1), in which TIPS-substituted DTBDT (DTBDT-TIPS) was selected as the D moiety, electron-withdrawn BT, 5,6-difluoro-BT (FBT) and NT units were selected as the A unit, and the long branched 2-octyldodecyl groups introduced the thiophene segment as conjugated π bridges for guaranteeing sufficient solution-processability. These components were designed and prepared to study the preliminary influence of the variation of the A unit on the photophysical, electrochemical, aggregation, and PV performances. The impacts of fluorination and the replacement of BT with NT on the absorption spectra, molecular energy levels, aggregation ability, backbone planarity, and mobility via tuning of the A moieties, resulted in the optical band gaps ($E_{g}^{\mathrm{opt}}$), *E*_HOMO_, and *E*_LUMO_, correspondingly varying in the ranges of 1.83~1.67 eV, −5.35~−5.46 eV, and −3.46~−3.66 eV, respectively. The absorption profile was slightly blue-shifted by the fluorination; however, it was greatly broadened after replacing the BT with NT, and the *E*_HOMO_ was deepened. It was revealed that PDTBDT-TIPS-DTNT-OD exhibited the best PCE of 3.37%, which was 1.29 times higher than its counterpart PDTBDT-TIPS-DTBT-OD, where such an improvement originated from the broadened absorption, lower *E*_HOMO_, and an optimized morphology. Unfortunately, the PCE decreased by 25.9% from fluorination, because of an oversized aggregation morphology and suppressed *J*_SC_, despite a high *V*_OC_ of 0.93 V.

## 2. Materials and Methods

### 2.1. Characterization

^1^H nuclear magnetic resonance (^1^H NMR) and ^13^C NMR spectra were measured on a Bruker DRX 600 (Rheinstetten, Germany) 600 MHz and 126 MHz spectrometer, respectively, with tetramethylsilane (TMS) as the internal reference. The chemical shifts were recorded in units of ppm and the splitting patterns were designed as s (singlet), d (doublet), t (triplet), m (multiplet), and br (broaden). Melting points were obtained on a microscopic melting point apparatus (Beijing Taike, Beijing, China), and the temperature gauge was uncorrected. C, H, and N elemental analyses (EAs) were carried out on a Vario EL Elemental Analysis Instrument (Elementar Co., Hanau, Germany). TGA curves were collected on TGA 2050 instruments (New Castle, DE, USA), at the heating rate of 10 °C min^−1^ and under an N_2_ flow rate (20 mL min^−1^). Polymer molecular weights were obtained using a Waters GPC 2410 in relative to polystyrene standards utilizing THF as the eluent. UV-Vis absorption measurement was performed on a UV-1800 spectrophotometer (Shimadzu, Kyoto, Japan). Thin film X-ray diffraction (XRD) was recorded on a PANalytical X′Pert PRO diffractometer (PANalytical Inc., Almelo, the Netherlands) equipped with a rotating anode (Cu Ka radiation, λ = 1.54056 Å). The electrochemical properties of the polymer films were measured on a CHI600D electro-chemical instrument (Shanghai Chenhua, Shanghai, China) in anhydrous CH_3_CN, at a scan rate of 100 mV s^−1^ under N_2_. Tetra(*n*-butyl)ammonium hexafluorophosphate (Bu_4_NPF_6_) (0.1 mol L^−1^) was utilized as the electrolyte. A three-electrode cell was used in all the experiments, wherein a glassy carbon electrode coated with polymer film, platinum wire, and an Ag/AgNO_3_ (0.01 mol L^−1^ of AgNO_3_ in CH_3_CN) electrode were used as the working, counter, and reference electrodes, respectively. The potential of the Ag/AgNO_3_ reference electrode was calibrated using the ferrocene/ferrocenium couple (Fc/Fc^+^), where the energy level was −4.80 eV. Note that the polymer thin films were obtained by drop casting 1 μL polymer chloroform solution, with a concentration of 1 mg mL^−1^, onto the glass carbon electrode, and then it was dried in air. Atomic force microscopy (AFM) images (5 × 5 μm^2^) were acquired on an MFP-3D-SA (Asylum Research, Santa Barbara, CA, USA) in a tapping mode. Transmission electron microscopy (TEM) images were acquired with a Tecnai G^2^ F20 (FEI, Hillsboro, OR, USA) transmission electron microscope at an accelerating voltage of 200 kV.

### 2.2. Materials

All reagents were purchased from commercial sources (Sigma-Aldrich (Shanghai), Shanghai, China; Acros, Belgium, USA; J&K, Beijing, China; and TCI (Shanghai), Shanghai, China), and were used as received without further purification, unless otherwise stated. THF and Et_2_O were distilled from sodium/benzophenone and were freshly distilled before use. The conjugated polyelectrolyte material PFN, which utilized was as an electron-interface layer as in Reference [64], and dibromide 4,9-bis(5-bromo-4-(2-octyldodecyl)thien-2-yl)naphtho[1,2-*c*:5,6-*c*′]bis[1,2,5]thiadiazole (DTNT-ODBr_2_) [65], were synthesized according to the reported methods. The synthetic routes of the dibromides 4,7-bis(5-bromo-4-(2-octyl-dodecyl)thien-2-yl)benzo[*c*][1,2,5]thiadiazole (DTBT-ODBr_2_), the 4,7-bis- (5-bromo-4-(2-octyldodecyl)thien-2-yl)-5,6-difluorobenzo[*c*][1,2,5]thiadiazole (DTFBT-ODBr_2_), and the bistin 2,7-bis(trimethyltin)-5,10-bis(triisopropylsilylethynyl)dithieno[2,3-*d*:2′,3′-*d*′]benzo[1,2-*b*:4,5-*b*′]-dithiophene (DTBDT-TIPSSn) as described in Reference [54], were seen in the Appendix A.

### 2.3. Polymer Synthesis

The general procedure adopted for polymer synthesis was as follows: Carefully purified bistin monomer DTBDT-TIPSSn and dibromo-monomer (DTBT-ODBr_2_, DTFBT-ODBr_2_, and DTNT-ODBr_2_) were dissolved into 6 mL degassed dry toluene and 0.8 mL DMF in a 25 mL two-neck round-bottom flask under argon (Ar) conditions. The mixture was bubbled with Ar for another 20 min to remove O_2_. Thereafter, Pd_2_(dba)_3_ (1.4 mg), P(o-tolyl)_3_ (2.3 mg) were quickly added to the mixture in one portion and the solution was bubbled with Ar for another 20 min. The mixture was then vigorously refluxed for 48 h under Ar, followed by the subsequent addition of 2-tri(butylstannyl)thiophene and 2-bromothiophene at an interval of 8 h for ending-capping. After further reflux at 8 h, the mixture was poured into 300 mL methanol. The precipitate was collected by filtration and the crude polymer was subjected to Soxhlet extraction successively with ethanol, acetone, hexane, and toluene. The toluene fraction was condensed to about 6 mL and precipitated into methanol. The black solid was collected and completely dried under vacuum overnight to obtain the target material with a yield of 56.9%~81.2%

#### 2.3.1. Poly[5,10-bis(triisopropylsilylethynyl)dithieno[2,3-*d*:2′,3′-*d*′]benzo[1,2-*b*:4,5-*b*′]dithiophene-2,7-diyl-*alt*-4,7-bis(4-(2-octyldodecyl)thien-2-yl)benzo[*c*][1,2,5]thiadiazole-5,5′-diyl] (PDTBDT-TIPS-DTBT-OD)

DTBDT-TIPSSn (100.9 mg, 0.102 mmol) and DTBT-ODBr_2_ (104.0 mg, 0.102 mmol) were used to prepare the PDTBDT-TIPS-DTBT-OD according to the general procedure illustrated above. A black solid of 101.6 mg was obtained with a yield of 65.5%. Number-average molecular weights (*M*_n_) = 13.4 kDa, polydispersity index (PDI) = 1.80. ^1^H NMR (600 MHz, CDCl_3_), *δ* (ppm), 8.03 (m, ArH), 7.84 (m, ArH), 7.62 (m, ArH), 7.44 (m, ArH), 7.07 (m, ArH), 2.91 (br, CH_2_), 1.97 (br, CH), 1.80–0.7 (m, CH, CH_2_, CH_3_). Anal. Calcd for C_90_H_130_N_2_S_7_Si_2_: C, 71.09%; H, 8.62%; N, 1.84%. Found, C, 71.00%; H, 8.50%; N, 1.95%.

#### 2.3.2. Poly[5,10-bis(triisopropylsilylethynyl)dithieno[2,3-*d*:2′,3′-*d*′]benzo[1,2-*b*:4,5-*b*′]dithiophene-2,7-diyl-*alt*-4,7-bis(4-(2-octyldodecyl)thien-2-yl)-5,6-difluorobenzo[*c*][1,2,5]thiadiazole-5,5′-diyl] (PDTBDT-TIPS-DTFBT-OD)

DTBDT-TIPSSn (97.0 mg, 0.098 mmol) and DTFBT-ODBr_2_ (103.0 mg, 0.098 mmol) were used to prepare the PDTBDT-TIPS-DTFBT-OD according to the general procedure illustrated above. A black solid of 86.8 mg was obtained with a yield of 56.9%. *M*_n_ = 14.4 kDa, PDI = 1.90. ^1^H NMR (600 MHz, CDCl_3_), *δ* (ppm), 8.15 (m, ArH), 7.93 (m, ArH), 7.61 (m, ArH), 7.50 (m, ArH), 7.37 (m, ArH), 7.13 (m, ArH), 2.88 (br, CH_2_), 1.90 (br, CH), 1.40–1.10 (m, CH, CH_2_), 0.77 (t, CH_3_). Anal. Calcd for C_90_H_1280_F_2_N_2_S_7_Si_2_: C, 69.44%; H, 8.291%; N, 1.80%. Found, C, 69.01%; H, 8.51%; N, 1.90%.

#### 2.3.3. Poly[5,10-bis(triisopropylsilylethynyl)dithieno[2,3-*d*:2′,3′-*d*′]benzo[1,2-*b*:4,5-*b*′]dithiophene-2,7-diyl-*alt*-4,9-bis(4-(2-octyldodecyl)thien-2-yl)naphto[1,2-*c*:5,6-*c*′]bis[1,2,5]thiadiazole-5,5′-diyl] (PDTBDT-TIPS-DTNT-OD)

DTBDT-TIPSSn (102.9 mg, 0.104 mmol) and DTNT-ODBr_2_ (117.4 mg, 0.104 mmol) were used to prepare the PDTBDT-TIPS-DTNT-OD according to the general procedure illustrated above. A black solid of 137.6 mg was obtained with a yield of 81.2%. *M*_n_ = 13.6 kDa, PDI = 1.94. ^1^H NMR (600 MHz, CDCl_3_), *δ* (ppm), 9.05 (m, ArH), 7.93 (m, ArH), 8.39 (br, ArH), 8.02 (m, ArH), 7.74 (m, ArH), 7.37 (m, ArH), 7.01 (m, ArH), 6.73 (m, ArH), 2.91 (br, CH_2_), 2.04 (br, CH), 1.40–1.10 (m, CH, CH_2_), 0.85 (m, CH_3_). Anal. Calcd for C_94_H_130_N_4_S_8_Si_2_: C, 69.32%; H, 8.04%; N, 3.44%. Found, C, 69.20%; H, 8.01%; N, 3.47%.

### 2.4. Fabrication and Characterization of the PSCs

A patterned indium tin oxide (ITO) coated glass with a sheet resistance of 10–15 Ω/square was sequentially cleaned using detergent, deionized water, acetone, and *iso*-propanol in an ultrasonic cleaner. Solar cell devices were fabricated using an inverted configuration of the ITO/PFN/active layer/MoO_3_/Ag. Note that the interlayer PFN layer was prepared according to the reported method described in Reference [65]. The active layer, with a thickness in the 90–110 nm range, was deposited on top of the interlayer layer by spin-casting from the chlorobenzene (CB) solution containing the studied copolymers/PC_61_BM (*w*/*w*; 1:1, 1:1.5, and 1:2), respectively. The thickness of the photosensitive layer was approximately 80–120 nm, detected on a surface profilometer (DektakXT, Bruker, MA, USA). A MoO_3_ layer with a thickness of about 8 nm was thermally evaporated at an evaporation rate of 0.1 Å s^−1^, at a vacuum degree of 5 × 10^−5^ Pa. At the end of the fabrication process, the Ag layer (100 nm) was evaporated by a shadow mask. The overlapping area between the cathode and anode was defined by a pixel size of 0.10 cm^2^. The thickness of the hole-transporting MoO_3_ layer and the metal Ag electrode, were both monitored using a quartz crystal thickness/ratio monitor (SI-TM206, Shenyang Sciens Co., Shenyang, China). All the fabrication processes were finished in a nitrogen drybox (Etelux Co., Beijing, China), where its contents of oxygen and moisture were both less than 1 ppm. The current density–voltage (*J*–*V*) curves were measured on a Keithley 2400 source-measurement unit. The PCEs of the PSCs were measured under a 1 sun, AM 1.5 G (Air mass 1.5 global) spectrum using a XES-70S1 (San-EI Electric Co. Ltd., Osaka, Japan) solar simulator (AAA grade), with an irradiation of 100 mW cm^−2^. The external quantum efficiency (EQE)/incident photon to charge carrier efficiency (IPCE) was obtained on a 7-SCSpecIII Solar Cell Spectral Response Measurement System (Beijing 7-star Opt. In. Co., Beijing, China), and the absolute photosensitivity was determined using a calibrated silicon detector.

### 2.5. Hole-Only Device Fabrication and Measurement

The hole mobility of the active layer was measured from the J–V curves obtained under dark current using the steady state space charge-limited current (SCLC) method, with the hole-only device configuration of the ITO/PEDOT:PSS/active layer/MoO_3_/Ag. The processing conditions for the active layers were the optimized one. The J–V curves were fitted using the Mott-Gurney square law as References [65,66]: $J=\frac{9}{8}\varepsilon_{0}\varepsilon_{r}\mu \frac{V^{2}}{L^{3}}$, where *J* is the current density, *ε*_0_ is the free space permittivity (8.85 × 10^−12^ F m^−1^), *ε*_r_ is the dielectric constant of the polymer (assumed to be 3), *μ* is the hole mobility, *L* is the thickness of the BHJ active layer film, and the effective potential *V* = (*V*_appl_ − *V*_bi_): where *V*_appl_ is the applied potential and *V*_bi_ is the built-in voltage resulting from difference in the work function of the anode and cathode. Finally, the mobility was extracted from the slope and *L* according to $\mu =\frac{{\mathrm{slope}}^{2}\times 8L^{3}}{9\varepsilon_{0}\varepsilon_{r}}$, by linearly fitting *J*^1/2^ with *V*.

## 3. Results

### 3.1. Molecular Design, Synthesis, and Characterization

Synthetic pathways for dibromide DTBT-ODBr_2_, DTFBT-ODBr_2_, and bistin comonomer DTBDT-TIPSSn [54], are outlined in Appendix A. The structures and purity of all the compounds were fully characterized and identified by ^1^H NMR (Appendix A) and elemental analysis. The DTNT-ODBr_2_ was synthesized according to our reported procedure in Reference [65]. As depicted in Scheme 1, three target CPs, PDTBDT-TIPS-DTBT-OD, PDTBDT-TIPS-DTFBT-OD, and PBDT-TIPS-DTNT-OD bearing different A units, were prepared using a typical Stille polymerization reaction between the bistin DTBDT-TIPSSn and the dibrominated DTBT-ODBr_2_, DTFBT-ODBr_2_, and DTNT-ODBr_2_ in toluene, with the use of tri(dibenzylidene-acetone)dipalladium(0) [Pa_2_(dba)_3_] as a catalyst and tri(*o*-tolyl)phosphine [P(*o*-tol)_3_] as a ligand as described in Reference [67]. Note that, to improve the stability of the copolymers, the end-capping was performed with 2-tributyl-stannylthiophene and 2-bromothiophene as in Reference [22]. The product polymers were purified by Soxhlet extractions using ethanol, acetone, hexane, and toluene as the eluent to remove oligomers and catalyst impurities. Ultimately, the toluene-soluble fraction was recovered by being reprecipitated in methanol and then dried under vacuum overnight to remove the residual solvents. These copolymers were obtained as black solids with yields of 56.9%~81.2%. PDTBDT-TIPS-DTBT-OD and PDTBDT-TIPS-DTNT-OD possessed good solubility in common organic solvents, such as chloroform (CF), CB, and *o*DCB at an ambient temperature. However, fluorinated PDTBDT-TIPS-DTBT-OD exhibited limited solubility, except for at an elevated temperature. The ^1^H NMR spectra of these studied copolymers are shown in Appendix A. *M*_n_ and PDIs values were estimated to be 13.4 kDa and 1.80 for PDTBDT-TIPS-DTBT-OD; 14.4 kDa and 1.90 for PDTBDT-TIPS-DTFBT-OD; and 13.6 kDa and 1.94 for PDTBDT-TIPS-DTNT-OD, respectively, Appendix A implies that the effect caused by different molecular weights could be ignored. The thermal properties of these copolymers were evaluated by thermogravimetry analysis (TGA) under a nitrogen atmosphere. As shown in Appendix A, the decomposition temperatures (*T*_d_) at a 5% weight-loss ranged from 352 to 387 °C (Appendix A), suggesting that the studied CPs possessed adequate thermal stability for the fabrication of PSCs.

### 3.2. Optical Property

The effect of the electro-withdrawing of fluorine onto BT and the replacement of BT with NT on absorption was investigated in both the CB solutions and as solid films. Normalized UV-Vis absorption spectra are elucidated in Figure 2, and the detailed parameters are summarized in Table 1. The absorption profile of PDTBDT-TIPS-DTBT-OD was similar to the analogue PTBDT-BT [17]. The absorption peaks at the high energy region are ascribed to the π–π* transition from the main chain units of the polymer backbone, while ones at low energy region are caused by the ICT transition [68]. Obviously, in the CB solution, the maximum absorption peak was blue-shifted by 17 nm from 552 to 535 nm by the fluorination, but a dramatic red-shift value of approximately 80 nm and one distinct shoulder peak were observed after incorporating NT into the polymer backbone, suggesting that there existed a stronger interaction even in the diluted solution as in Reference [25]. Going from solution to film state, the red-shifted values of the maximum absorption peaks were 30, 47, and 3 nm for the PDTBDT-TIPS-DTBT-OD, PDTBDT-TIPS-DTFBT-OD, and PBDT-TIPS-DTNT-OD, respectively. We also observed that there existed weaker shoulder peaks at about 620 nm for the former two BT-based CPs, and the blue-shifted value of 14 nm of the shoulder peak for PBDT-TIPS-DTNT-OD. The optical bandgaps ($E_{g}^{\mathrm{opt}}$) of the films for the copolymers could be estimated using the absorption edge ($\lambda_{\mathrm{onset}}^{\mathrm{film}}$), according to the empirical equation $E_{g}^{\mathrm{opt}}$ = 1240/$\lambda_{\mathrm{onset}}^{\mathrm{film}}$, and accordingly the $E_{g}^{\mathrm{opt}}$s were calculated as 1.82, 1.83, and 1.67 eV for the PDTBDT-TIPS-DTBT-OD, PDTBDT-TIPS-DTFBT-OD, and PBDT-TIPS-DTNT-OD, respectively. Plainly, the $E_{g}^{\mathrm{opt}}$ was almost unaffected by the fluorination, but it was effectively reduced by replacing the BT with NT. Apart from the absorption profile, the absorption coefficient was also a pivotal parameter in judging the ability to harvest solar light of CPs [69]. Therefore, the corresponding absorption coefficients in both the CB (Appendix A) and film (Appendix A) were tested, and PBDT-TIPS-DTNT-OD exhibited the highest coefficient at the maximum absorption in film ($\varepsilon_{\max}^{\mathrm{film}}$), and the fluorination decreased the corresponding absorption coefficients (Appendix A). It was inferred that NT-based CP could harvest more sunlight which was propitious to achieve a good *J*_SC_.

It has been confirmed that with rising temperature the molecule motions of CPs are bound to be accelerated, resulting in a reduction of the absorption shoulder peaks with regard to the aggregation evoked by intermolecular π–π* transition. Namely, the distortion of the conjugated backbone is strengthened at the elevated temperature, leading to a decrease of the effective conjugation length of the polymer backbone, and hence the absorption peaks engendered by the localized π–π* transitions, and the ones emerged from ICT transitions are both blue-shifted [49,70]. To gain insight into the difference between fluorination and the replacement of BT with NT on the interchain π–π stacking in solution, temperature-dependent absorption (TD-Abs) spectra of these copolymers in CB solution were measured, as shown in Figure 3. It was exhibited that as the temperature increased from 25 °C to 105 °C, the ICT absorption peaks (0–0 peak for the BT-based CPs and 0–1 peak for the NT-based CP) were all blue-shifted and the absorption strength decreased. In detail, the blue-shifted values (Δλ) and the decreased absorption strength (ΔA) were 20 nm and 3.88% for PDTBDT-TIPS-DTBT-OD; 15 nm and 3.20% for PDTBDT-TIPS-DTFBT-OD; and 62 nm and 7.47% for PDTBDT-TIPS-DTNT-OD, respectively. Furthermore, the absorption peak (0–0 peak) related to the aggregation was only observed in the PDTBDT-TIPS-DTNT-OD and it showed a trend of blue-shifting and decreasing, and it completely disappeared until the temperature was heated to 65 °C. As observed from the results, the order of aggregation ability in the CB solution should be: PDTBDT-TIPS-DTNT-OD > PDTBDT-TIPS-DTFBT-OD > PDTBDT-TIPS-DTBT-OD [70].

### 3.3. X-Ray Diffraction (XRD) Analysis

To get a deeper perspective on the impact of fluorination and the substitution of BT with NT on intermolecular interaction in the film state, X-ray diffraction (XRD) was adopted to investigate the formation of ordered structures within these studied polymers. The preparation conditions for these pristine polymer films was casted from polymer CB solution onto glass substrate. As illustrated in Figure 4, all copolymers exhibited two discernible peaks. The first peaks in a small angle region were located at 2θ of 4.00° for PDTBDT-TIPS-DTBT-OD, 4.08° for PDTBDT-TIPS-DTFBT-OD, and 4.25° for PDTBDT-TIPS-DTNT-OD, such that the distance of the polymer backbone separated by the flexible side was 22.06, 21.63, and 20.77 Å, respectively, based on the Bragg’s law (i.e., λ = 2dsinθ) [71]. The broad diffraction peaks at the wide-angle region, which reflects the π–π stacking distance, were located at 2θ of 20.62° for PDTBDT-TIPS-DTBT-OD, 2θ of 20.91° for PDTBDT-TIPS-DTFBT-OD, and 23.25° for PDTBDT-TIPS-DTNT-OD, respectively, corresponding to π–π stacking distances of 4.30 Å, 4.24 Å, and 3.89 Å. It was inescapably clear that the order of aggregation in the solid film state was PDTBDT-TIPS-DTNT-OD > PDTBDT-TIPS-DTFBT-OD > PDTBDT-TIPS-DTBT-OD, which was in accordance with the previous one in the CB solution.

### 3.4. Electrochemical Property

It is well known that the *E*_HOMO_, *E*_LUMO_, and electrochemical bandgap ($E_{g}^{\mathrm{ec}}$) of as-prepared CPs were important parameters in photovoltaic application. Therefore, the cyclic voltammetry (CV) method was applied to characterize the electrochemical properties, and the *E*_HOMO_ and *E*_LUMO_ were estimated from the oxidation onset potential ($\varphi_{\mathrm{ox}}^{\mathrm{onset}}$) and the reduction onset potential ($\varphi_{\mathrm{red}}^{\mathrm{onset}}$), respectively. The CV curves were recorded in Figure 5a, and the detailed data were listed in Table 1. Three CPs exhibited the quasi-reversible p-doping and n-doping processes, which were important for p-type semiconductor materials. We could see that the $\varphi_{\mathrm{ox}}^{\mathrm{onset}}$s for PDTBDT-TIPS-DTBT-OD, PDTBDT-TIPS-DTFBT-OD, and PDTBDT-TIPS-DTNT-OD were observed at around 0.67, 0.78, and 0.75 V, respectively. Note that the reference electrode was calibrated using the ferrocene–ferrocenium (Fc/Fc^+^) redox couple, which had redox potential with an absolute energy level of −4.80 eV relative to the vacuum energy level [72]. Meanwhile, this potential *φ*_1/2_ of the Fc/Fc^+^ redox couple was found to be 0.12 V under the exact same conditions. Accordingly, the *E*_HOMO_ and *E*_LUMO_ were calculated from the following equation: *E*_HOMO_ = −e($\varphi_{\mathrm{ox}}^{\mathrm{onset}}$ + 4.68) (eV) and *E*_LUMO_ = −e($\varphi_{\mathrm{red}}^{\mathrm{onset}}$ + 4.68) (eV), where $\varphi_{\mathrm{ox}}^{\mathrm{onset}}$ and $\varphi_{\mathrm{red}}^{\mathrm{onset}}$ are the onset oxidation and/or the reduction potentials vs. Ag/AgNO_3_, and the corresponding *E*_HOMO_/*E*_LUMO_ values were approximately −5.35/−3.46 eV, −5.46/−3.54 eV, and −5.43/−3.66 eV for PDTBDT-TIPS-DTNT-OD, PDTBDT-TIPS-DTFBT-OD, and PBDT-TIPS-DTNT-OD, respectively. Clearly, the depressed *E*_HOMO_ values were 0.11 and 0.08 eV for fluorination and the replacement BT with NT, respectively, which were both prone to obtaining high a *V*_OC_ in the PSCs [4]. The $E_{g}^{\mathrm{ec}}$ was calculated from the equation $E_{g}^{\mathrm{ec}}$ = e($\varphi_{\mathrm{ox}}^{\mathrm{onset}}$ − $\varphi_{\mathrm{red}}^{\mathrm{onset}}$) (eV), and the $E_{g}^{\mathrm{ec}}$ values were 1.89, 1.92, and 1.77 eV for DTBDT-TIPS-DTNT-OD, PDTBDT-TIPS-DTFBT-OD, and PBDT-TIPS-DTNT-OD, respectively. These observed values were slightly higher (0.07–0.10 eV) than those of the $E_{g}^{\mathrm{opt}}$, which presumably resulted from the exciton binding energies of the CPs and/or the interfacial barriers for charge injection [73]. For a better comparison, the energy levels diagram of the three donor copolymers and PC_71_BM is described in Figure 5b. It could be inferred that all the polymers were suitable for use as donor materials to match well with the PC_71_BM, which had LUMO gaps of 0.54~0.74 eV, exhibiting sufficient driving force to facilitate an exciton dissociation at the D–A interfaces, thereby guaranteeing energetically favorable electron transfer to overcome the binding energy of the intrachain exciton [11]. Interestingly, the replacement of BT with NT could simultaneously achieve the balance of deepening the *E*_HOMO_ and reducing the bandgap of the donor polymer, which differed from the observations in PBDT-DTNT [25] and PBDTT-TNAT [11]. This meant that an obviously decreased bandgap of the PDTBDT-TIPS-DTNT-OD principally benefited from the remarkably deepened *E*_LUMO_.

### 3.5. Theoretical Calculation

To further inspect the effect of fluorination and replacement of the BT with NT on the molecular backbone conformation and the electron density distributions, the calculations were carried out the using density functional theory (DFT) calculation on the B3LYP/6-31G* basis set, implemented in the Gaussian 09 program suite as in Reference [74]. To save time and cost, the long-branched OD side chain and TIPS groups were substituted by methyl and trimethylsilylethynyl, and the corresponding polymer backbones were simplified into oligomers with one repeating units (unimer), prior to the calculations. As presented in Figure 6, the HOMO orbitals are delocalized across the whole conjugated main chain, whereas, the LUMO orbitals are preferentially concentrated in the corresponding electron-deficient segments. These DFT calculations implied that the effective charge-transfer process would occur between the DTBDT-TIPS and BT/FBT/NT units. Accordingly, the calculated *E*_HOMO_, *E*_LUMO_, and bandgap were −4.97, −2.68, and 2.29 eV for PDTBDT-TIPS-DTBT-OD; −5.04, −2.78, and 2.26 eV for PDTBDT-TIPS-DTFBT-OD; and −5.01, −2.94, and 2.07 eV for PDTDBT-TIPS-DTNT-OD, respectively. Clearly, the variation trend of the *E*_HOMO_ was deepened by the fluorination and replacement of the BT with NT, agreeing well with results from the previous CV measurements. Moreover, it has already been demonstrated that good planarity of the polymer backbone is conducive to ensuring closer π–π stacking, which is propitious to internal aggregation, charge transfer, and molecular conjugation to engender a lower band gap [2]. Consequently, the dihedral angles *θ*_1_ between the donor DTBDT-TIPS unit and the alkylthiophene bridge, *θ*_2_, and *θ*_3_, between the alkylthiophene bridge and the acceptor BT/FBT/NT moiety were 21.79°, −7.24°, and −4.10° for the DTBDT-TIPS-DTBT-OD; 22.29°, −1.06°, and −0.17° for the DTBDT-TIPS-DTFBT-OD; and 15.65°, −8.78°, and 2.96° for the DTBDT-TIPS-DTNT-OD, respectively, as shown in Appendix A. The molecular planarity was improved by both the fluorination and replacement of BT with NT, in good accordance with the previous XRD analyses and TD-Abs measurements. Interesting to note that the calculated data were all higher in energy than those from the experimental test, since in the calculation process only the finite unimer and the short side chain were considered in the DFT model to describe the conjugated polymers. Notwithstanding, these theoretical calculation values agreed with the trends observed in the experiment and still disclosed the interrelation amongst the molecular structure, optoelectrical property, and thus the PV performance.

### 3.6. Photovoltaic Properties

To investigate the PV properties of these copolymers as donor materials in BHJ PSCs, the inverted devices evoked significantly the enhanced device stability where a structure of the ITO/PFN/polymers:PCBM/MoO_3_/Ag was fabricated. Here, the active layers consisted of copolymers, and PC_61_BM or PC_71_BM were spin-coated from the CB solution, whilst a 10 nm thick PFN and an 8 nm thick MoO_3_ were utilized as the cathode and anode interlayer, respectively, as described in References [75,76]. The performance was optimized using various processing parameters, including the D/A blend ratio, the usage of processing additive, and the replacement of PC_61_BM with PC_71_BM. The detailed device fabrication process is presented in the Appendix A). *J*–*V* characteristics were characterized under AM 1.5G illumination at 100 mW cm^−2^ using a solar simulator. Firstly, it is widely recognized that the donor polymer/PC_61_BM (D/A) weight ratio plays a significant and vital role in improving the PV performance of the corresponding devices [49,77,78]. Consequently, the devices from each of the polymer weight ratios of the D/A ratios, such as 1:1, 1:1.5, to 1:2 were fabricated, and it was found that the best D/A ratio for all the copolymers was 1:1.5 in Appendix A, and the detailed parameters are summarized in Appendix A. The PDTBDT-TIPS-DTBT-OD-based device showed the best PCE of 1.37%, with a *V*_OC_ of 0.91 V, a *J*_SC_ of 3.32 mA cm^−2^, and a *FF* of 45.21%. The fluorinated PDTBDT-TIPS-DTFBT-OD exhibited an inferior PCE of 1.09%, with a lower *J*_SC_ of 2.19 mA cm^−2^, even with an increased *V*_OC_ of 0.93 V, and *FF* of 53.59%. The PDTBDT-TIPS-DTNT-OD-based device also showed an inferior PCE of 1.15%, with a simultaneous decreased *V*_OC_ of 0.88 V, a *J*_SC_ of 3.19 mA cm^−2^, and a *FF* of 40.55%. Obviously, the EQE curves in Appendix A validated the variations of the *J*_SC_ in *J*–*V* measurement.

Plenty of researches have demonstrated that the solvent additive 1,8-diiodoctane (DIO) can help active layers to form a more ordered and nanoscale bicontinuous interpenetration network, which facilitates efficient charge generation and transport, and hence improves the PV performance of the corresponding devices [79,80]. Subsequently, the 3% DIO (DIO/CB, V/V) was selected as a solvent additive during the device fabrication and the *J*–*V* curves are shown in Appendix A. A significant 163% enhancement in the PCE (from 1.15% to 3.03%) in the PDTBDT-TIPS-DTNT-OD-based device was achieved, which benefited from a large 120% enhancement in the *J*_SC_ (from 3.19 to 7.03 mA cm^−2^) and a 21.8% increase in the *FF* (from 40.55% to 49.39%), even though the *V*_OC_ remained stable. For the control PDTBDT-TIPS-DTBT-OD-based device, it exhibited a slightly inferior PCE of 1.03%, influenced by a decline of 43% in the *J*_SC_ (from 3.32 to 1.89 mA cm^−2^). Unfortunately, the fluorinated PDTBDT-TIPS-DTFBT-based device showed a 59.6% drop in the PCE (from 1.09% to 0.44%), which was ascribed to simultaneous 3.2%, 53.0%, and 26.1% declines in the *V*_OC_, *J*_SC_, and *FF* (from 0.93 to 0.90 V, from 2.19 to 1.03 mA cm^−2^, and from 53.59% to 39.58%), respectively. Evidently, the DIO may play a positive role in the PDTBDT-TIPS-DTNT-OD-based cell; however, it gives rise to an adverse effect in the PDTBDT-TIPS-DTBT-OD- and PDTBDT-TIPS-DTFBT-OD-based devices.

Owing to PC_71_BM bearing similar electronic properties to PC_61_BM, but with a considerably higher absorption coefficient and a broader absorption spectrum, PC_71_BM was further applied instead of PC_61_BM to maximize the PCE [81]. As seen in Appendix A, the PCE of the PDTBDT-TIPS-DTBT-OD based device slightly increased by approximately 7.3% (from 1.37% to 1.47%), which originated from a 9.0% enhancement in the *J*_SC_ (from 3.32 to 3.62 mA cm^−2^) and the *FF* (from 45.21% to 50.93%), but with a decreased *V*_OC_ (from 0.91 V to 0.80 V). Regrettably, an improvement in the PCE for the fluorinated system was not observed. However, as for the PDTBDT-TIPS-DTFBT-OD-based device, an 11.2% enhancement in the PCE (from 3.03% to 3.37%) was observed, and this improvement was ascribed to the 2.56% (from 7.03 to 7.21 mA cm^−2^) and 3.60% (from 49.39% to 52.99%) increases in the corresponding *J*_SC_ and *FF*, whilst the *V*_OC_ remained stable. Evidently, these variations agreed well with the corresponding EQE curves in Appendix A, in particular, there existed a distinct improvement ranging from 400 to 550 nm in the EQE profile related to the contribution of the PC_71_BM.

Going through the previous series of device fabrication optimization, including the D/A ratio, the DIO additive, and the replacement of PC_61_BM with PC_71_BM, the best *J*–*V* curves and EQE spectra of the studied polymers are shown in Figure 7, and the best PV data are listed in Table 2. Owing to incorporation of the DTBDT-TIPS into the polymer backbones as D units, the polymers containing the TIPS PV devices all had a higher *V*_OC_ in the range of 0.80–0.93 V, and the *V*_OC_ values increased after fluorination and the replacement of the BT with NT, in agreement with the predictions of a previous deeper *E*_HOMO_ in the DFT calculation and CV measurement [4]. It was exhibited that the optimal PCE decreased from 1.47% to 1.09% with the incorporation of fluorine into BT, which was ascribed to the 39.5% reduction in the *J*_SC_ (from 3.62 to 2.19 mA cm^−2^), despite a 16.25% increase in the *V*_OC_ (from 0.80 to 0.93 V) and a 5.16% increase in the *FF* (from 50.96% to 53.59%). Interestingly, when replacing the BT with NT, a 129% enhancement in the PCE (from 1.47% to 3.37%) was realized, which benefited from 10% (from 0.80 V to 0.88 V), 99.2% (from 3.62 to 7.21 mA cm^−2^), and 3.98% (from 50.96% to 52.99%) increases in the corresponding *V*_OC_, *J*_SC_, and *FF* of the devices. From Figure 7b, photo-response profiles of the optimal devices in the ranges of 300–720 nm for the PDTBDT-TIPS-DTBT-OD, 300–680 nm for the PDTBDT-TIPS-DTFBT-OD, and 300–760 nm for the PDTBDT-TIPS-DTNT-OD, were observed. The higher *J*_SC_ in the PDTBDT-TIPS-DTNT-OD-based device was due to its broadened absorption and higher EQE value. Moreover, the integrated *J*_SC_ values from the EQE curves were 3.50, 2.08, and 7.09 mA cm^−2^, for the best-performing PDTBDT-TIPS-DTBT-HD:PC_71_BM, PDTBDT-TIPS-DTFBT-HD:PC_61_BM, and PBDT-TIPS-DTNT-OD:PC_71_BM, respectively, which was indicative of an error smaller than 5% compared to the *J*_SC_ value obtained from the *J*–*V* curves.

### 3.7. Charge Mobilities

To find the reason for the different photovoltaic performance on fluorination and on replacement of the BT with NT on the polymer backbone, the vertical hole transport property was examined and hole-only devices of polymer:PCBM blend films were fabricated under identical conditions to the optimized PSCs. The hole mobilities estimated using the SCLC method could be described by the equation $J=\frac{9}{8}\varepsilon_{0}\varepsilon_{r}\mu_{h}\frac{V^{2}}{L^{3}}$ as described in References [66,82]. Worth noting that the thickness was 87 nm for the PDTBDT-TIPS-DTBT-OD, 80 nm for the PDTBDT-TIPS-DTFBT-OD, and 127 nm for the PDTBDT-TIPS- DTNT-OD, respectively. As shown in Figure 8, the *J*^1/2^–*V* curves of the active layers were obtained in the dark for the hole-only devices. The perfect linear fitting in the figure indicates that the *J*^1/2^–*V* follow the Mott-Gurney square law. Accordingly, the *μ*_h_s of the PDTBDT-TIPS-DTBT-OD, PDTBDT-TIPS- DTFBT-OD, and PDTBDT-TIPS-DTNT-OD were calculated to be 2.32 × 10^−5^, 3.98 × 10^−5^, and 9.09 × 10^−5^ cm^2^ V^−1^ s^−1^, respectively, (Appendix A). Apparently, such variation tendencies in mobility was consistent with the results of the XRD and DFT calculations. The highest SCLC hole mobility could partially account for the observed markedly higher *J*_SC_ in the PDTBDT-TIPS-DTNT-OD-based cell.

### 3.8. Film Morphology

As is known, performances of BHJ PSCs are strongly associated with the morphologies of the active layer [79,83]. Therefore, to understand the reasons why the fluorination and the NT substitution of BT had such different photovoltaic performances, the morphologies of the optimized blend films were investigated using tapping-model atomic force microscopy (AFM) on a surface area of 5 × 5 μm^2^. The AFM topography and the phase images of the three copolymers showed the different morphologies. As shown in Figure 9, the blend film of the PDTBDT-TIPS-DTBT-OD:PC_71_BM with 3% DIO had a very smooth surface with a root-mean-square (RMS) roughness of only 0.478 nm, but with an inconspicuous phase separation. For the blend film of the fluorinated PDTBDT-TIPS-DTFBT-OD:PC_61_BM without DIO, it exhibited a very rough surface with an RMS of 10.62 nm and oversized aggregation, which severely limits the effective exciton diffusion and deteriorates the exciton dissociation probability by reducing the D/A interfacial areas, resulting in a poor *J*_SC_ [83]. Worth noting that the blend film of PDTBDT-TIPS-DTNT-OD:PC_71_BM with 3% DIO had an appropriate surface with an RMS roughness of 5.202 nm and a proper phase separation, forming a bicontinuous interpenetrating network structure, which could support the D/A interfacial areas to promote the exciton dissociation into the free charges, but it also assisted the free charges to transport to the corresponding electrodes [4]. To further inspect the composition and in-depth morphology of the active layers, transmission electron microscopy (TEM) was applied. As is known, owing to the big difference in electron scattering density between polymers (1.1 g cm^−3^) and fullerene derivatives (1.5 g cm^−3^), the polymer networks and porous regions are imaged as lighter (copolymer) and darker (PC_71_BM) colored in the TEM, respectively [65]. As depicted in Figure 10, it was found that after incorporating F into BT, there was oversized polymer aggregation. Meanwhile, the improved phase separation and partial fibril nanostructure were observed when substituting the BT with NT, similar to the results observed in former AFM. Therefore, the morphological information could partially explain that the fluorination led to a decreased *J*_SC_, whereas the replacement of the BT with NT resulted in a significantly increased *J*_SC_ and slightly improved *FF*.

## 4. Conclusions

To sum up, three alternated D-π-A type CPs, PDTBDT-TIPS-DTBT-OD, PDTBDT-TIPS- DTFBT-OD, and PDTBDT-TIPS-DTNT-OD, in which A units were varied among BT, FBT, and NT, were designed and prepared. By changing the A moieties of the D-π-A polymer backbone, $E_{g}^{\mathrm{opt}}$, *E*_HOMO_ and *E*_LUMO_ could be validly tuned and varied in the ranges of 1.83~1.67 eV, −5.35~−5.46 eV, and −3.46~−3.66 eV, respectively. It was exhibited that the absorption profile was scarcely influenced by the fluorination on BT; however, it was greatly broadened after replacing the BT with NT. The *E*_HOMO_s were effectively deepened and the intermolecular aggregation in the CB solution and solid film state were improved. The optimal photovoltaic measurements demonstrated that replacing the BT with NT in PDTBDT-TIPS-DTNT-OD exhibited a simultaneous elevated *V*_OC_ of 0.88 V, *J*_SC_ of 7.21 mA cm^−2^, and *FF* of 52.99%, and resulted in a PCE of 3.37%, which was 1.29 times higher relative to its counterpart. These improvements originated from the greatly broadened absorption, lower *E*_HOMO_, improved molecular ordered structure, and the enhanced SCLC mobility and favorable morphology of the active layer. However, the PCE decreased by 25.9% from fluorination because of the suppressed *J*_SC_, resulting in an unfavorable blend film morphology despite a high *V*_OC_. Our preliminary results suggested that replacing the BT with NT in the D-π-A type polymer backbone was an effective method aimed at highly efficient PSCs.

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
