# Peer review of "Enhanced Photovoltaic Performance in D-π-A Copolymers Containing Triisopropylsilylethynyl-Substituted Dithienobenzodithiophene by Modulating the Electron-Deficient Units"

_polymers, 2018, doi:10.3390/polym11010012_

Round 1

Reviewer 1 Report

In the present paper, the authors reported enhanced photovoltaic performance in D-π-A copolymers containing triisopropylsilylethynyl-substituted Dithieobenzodithiophene. The results demonstrate that replacing BT with NT in D-π-A type polymer backbone is an effective strategy of tuning molecular structure for achieving high efficient PSCs.

View from the results of this work, this paper deserves the publication to polymers.

Author Response

Reviewer #1: In the present paper, the authors reported enhanced photovoltaic performance in D-π-A copolymers containing triisopropylsilylethynyl-substituted Dithienobenzodithiophene. The results demonstrate that replacing BT with NT in D-π-A type polymer backbone is an effective strategy of tuning molecular structure for achieving high efficient PSCs.

View from the results of this work, this paper deserves the publication to polymers.

Response: Thank you very much for your invaluable advice. We have carefully read revised the manuscript according to other two reviewer’s comments.

Reviewer 2 Report

The paper is well written and very complete. All given informations are of great interest (even SI).

Here are a few things to correct (minor revision):

-> a few spelling mistakes were found in chemical names 'dithieo' instead of 'dithieno', 'siyl' instead of 'silyl', 'bisitin' instead of 'bistin'

-> a sentence sound not english in line 26

-> why LUMO levels where not detected via electrochemistry reduction peaks? Explain the problem, or give the value.

-> line 55 : some additional references could be added like "Synthetic Metals 162 (2012) 1037– 1045"

Author Response

(1) A few spelling mistakes were found in chemical names 'dithieo' instead of 'dithieno', 'siyl' instead of 'silyl', 'bisitin' instead of 'bistin'.

Response: Thank you very much for your important suggestions. We have carefully revised our manuscript.

In line 4, 38, and 815, “Dithieobenzodithiophene” has been revised by “Dithienobenzodithiophene”.

In line 17, 115, 180, 196, 205, 214, 397, 865 and 869, “siyl” has been revised by “silyl”.

In line 868, “bisitin” has been revised by “bistin”.

(2) A sentence sound not english in line 26

Response: In line 26, the sentence “However, there existed the difference that little changed absorption in fluorination but greatly broadened absorption profile when replacing NT with BT.” has been changed into “However, there existed the difference that the absorption profile was little changed upon fluorination but greatly broadened when replacing BT with NT.”

(3) Why LUMO levels where not detected via electrochemistry reduction peaks? Explain the problem, or give the value.

Response: Thank you very much for your invaluable suggestions. According to your advice, we have tried our best to measure the reduction process of the studied CPs, and the results as shown in following diagram, and the calculated ELUMO values were –3.46 eV, –3.54 eV and –3.66 eV for PDTBDT-TIPS-DTNT-OD, PDTBDT-TIPS-DTFBT-OD, and PBDT-TIPS-DTNT-OD, respectively and electrochemical bandgap () of as-prepared CPs were 1.89, 1.92 and 1.77 eV for DTBDT-TIPS-DTNT-OD, PDTBDT-TIPS-DTFBT-OD, and PBDT-TIPS-DTNT-OD, respectively. These observed  values were slightly higher (0.07 ~0.10 eV) than , which presumably resulted from the exciton binding energies of the CPs and /or the interfacial barriers for charge injection (Chen, H.-C.; Chen, Y.-H.; Liu, C.-H.; et al., Polym. Chem., 2013, 4, 3411–3418; Wu P.-T.; Kim, F. S.; Champion R. D.; et al., Macromolecules 2008, 41, 7021–7028; Wang, Z.; Zhao, J.; Li, Y.; Polym. Chem., 2014, 5, 4984–4992). The corresponding changes have been added in my revised manuscript.

Table 1 Optical and electrochemical characteristics of copolymers.

Polymer

Solution

Film

1

(eV)

(V)

(V)

EHOMO2

(eV)

ELUMO3

(eV)

4

(eV)

λmax

(nm)

λsh

(nm)

λmax

(nm)

λsh

(nm)

λonset

(nm)

PDTBDT-TIPS-

DTBT-OD

366,425,552

375,582

620

680

1.82

0.67

1.22

–5.35

–3.46

1.89

PDTBDT-TIPS-

DTFBT-OD

365,427,535

375,582

622

678

1.83

0.78

1.14

–5.46

–3.54

1.92

PDTBDT-TIPS-

DTNT-OD

361,402,632

692

363, 405,

492, 635

678

740

1.67

0.75

1.02

–5.43

–3.66

1.77

1Optical band gap determined from the UV-vis absorption onset of the film ( = 1240/).

2Calculated from oxidation potential of the copolymer (EHOMO = –e( + 4.68) (eV)).

3Calculated from reduction potential of the copolymer (ELUMO = –e( + 4.68) (eV).

4Calculated from the equation  = e( ) (eV).

Figure 5 CV curves (a) and energy levels schematic diagram (b) of copolymers.

(4) Line 55 : some additional references could be added like "Synthetic Metals 162 (2012) 1037–1045"

Response: Thank you very much for your good advice. The suggested reference “Murugesan, V.; de Bettignies, R.; Mercier, R.; Guillerez, S.; Perrin, L. Synthesis and characterizations of benzotriazole based donor-acceptor copolymers for organic photovoltaic applications. Synthetic Met. 2012, 162, 10371045.” has been added in the revised manuscript (reference No. [69]).

Reviewer 3 Report

In this manuscript, Tong and Xia and their team demonstrated the important role of acceptor (A) building block in determining the photovoltaic performance of polymer solar cells. In their study, they prepared three conjugated polymers by using TIPS-substituted DTBDT (DTBDT-TIPS) as the donor moiety and choosing three A building blocks including electron-withdrawn BT, 5,6-difluoro-BT (FBT), and NT which has an extended conjugation. The influence of changing A unit on the photophysical, electrochemical, morphological, and photovoltaic performances were systematically investigated. It was shown that the absorption profile was barely changed upon fluorination. However, an obvious broadening absorption was observed when BT was replaced with NT. Most, impressively, NT based polymer exhibited the highest power conversion efficiency of 3.37%, which is much higher than those from BT based polymer and fluorinated polymer.

In general, this manuscript is one of the best papers I have reviewed for Polymers. The experiments and theoretical simulations are both excellently designed and executed. The data are sufficient and solid to support their conclusions. The presentation of the data is very clear to follow. In addition, the writing is smooth and easy to be understood by general audience in polymer science. Therefore, I believe that the impact of this paper will be high and would recommend acceptance of this manuscript to Polymers. But still there are some comments for the authors to address to further improve the manuscript. Please see below.

Comments

1.       Did the authors consider the effect of broad MW distribution on the PV performance? The authors employed traditional Stille polymerization to make the polymers. While the molecular weights of all polymers are similar. The PDIs of them are quite high in the range of 1.8 to 1.9.

2.       Following the first comment, did the author consider using controlled polymerization methods (such as catalyst-transfer polycondensation) to decrease the PDI of polymers?

3.       In page 2, line 73, please add “into” after “incorporated”.

4.       In page 4, line 147, “was” should be “were”.

5.       In page 4, line 171, “was” should be “were”.

6.       Did authors use SEM to characterize film morphologies as a supplementary to AFM?

Author Response

(1). Did the authors consider the effect of broad MW distribution on the PV performance? The authors employed traditional Stille polymerization to make the polymers. While the molecular weights of all polymers are similar. The PDIs of them are quite high in the range of 1.8 to 1.9.

Response: First, we want to express our thanks to your important and invaluable questions. In the polymer solar cell, the molecular weights and the distribution of the conjugated copolymers have an important impact on the solution processability, thermal stability, absorption, charge mobility and thus photovoltaic performance. In our former study, we found that with the increase of molecular weight, the maximum peak in the absorption would red-shifted and thus improved the short-circuit current density (J. Macromol. Sci. A 2017, 54, 3, 176–185). Besides, Yang et al. investigated a conjugated polymer with different molecular weight but similar PDI, found H-P4TFBT with high molecular weight exhibited better thermal stability and compatibility with PCBM, thus achieving a better PV property (Polym. Chem., 2015, 6, 6050–6057). Ding et al studied the molecular effect of PTB7 and found that molecular weight have a significant influence on PCE of corresponding devices. (J. Mater. Chem. A, 2016, 4, 7274–7280). In our study, to investigate the effect of the electron-withdrawing unit in D-π-A conjugated copolymers on optoelectronic properties, the molecular weights should not be ignored.

(2). Following the first comment, did the author consider using controlled polymerization methods (such as catalyst-transfer polycondensation) to decrease the PDI of polymers?

Response: Thank you very much for your valuable suggestions. In general, the semiconducting conjugated polymers were prepared by the methods, such as Stille coupling reaction between bistin and dibromide/diiodide comonomers, Suzuki coupling between diborate ester and dibromide/diiodide comonomers etc. The molecular weight and PDI values were mainly influenced by the purity of comonomers, solvents, reaction time, the utilized catalyst and heating methods (heating via oil-bath or assisted by microwave) and so on (Xia et al, J. Mater. Chem. A, 2014, 2, 15316–15325). The suggested method tuning PDI of CPs (catalyst-transfer polycondensation) is very good and we will adopt it to improve our conjugated polymers for polymer solar cell and further boost the PCE. Finally, thank you very much again.

(3). In page 2, line 73, please add “into” after “incorporated”.

Response: Thank you very much for your good advice. The “incorporated” has been changed to “incorporated into” in line 73 of page 2.

(4). In page 4, line 147, “was” should be “were”.

Response: Thanks for your kind advice. Due to this part has been rewritten, this error has been corrected in our new manuscript. Besides, in line 77, “an facile and effective strategy …” has been replaced by “an effective and facile strategy …”.

(5). In page 4, line 171, “was” should be “were”.

Response: Thank you very much for your kind advice. This error has been corrected in the revised manuscript (Line 173).

(6). Did authors use SEM to characterize film morphologies as a supplementary to AFM?

Response: Thank you very much for your constructive suggestions on the morphology study. As known, AFM only can examine the surface morphology information on the blend films which can be obtain the roughness and aggregation of the blend film, while transmission electron microscopy (TEM) can directly determine the composition and more real morphology of the blend film (Li Y., Chen Y., Liu X., et al. Macromolecules, 2011, 44, 6370–6381; Li G., Zhao B., Kang C., et al. 2015, ACS Appl. Mater. Interfaces, 2015, 7, 10710–10717; Liu X., Liu T., Duan C., et al. J. Mater. Chem. A, 2017, 5, 1713–1723). Therefore, we have further studied the in-depth morphology of the active layers using TEM, as shown in Fig. 10 in the revised manuscript. It was found that that after incorporating F into BT there existed the oversized polymer aggregation, meanwhile, the improved phase separation and even partial fibril nanostructure were observed when BT was substituted with NT, these variation tendencies were similar with the results observed in AFM.

Figure 10 TEM bright field images of active layers of PDTBDT-TIPS-DTBT-OD/PC71BM (1:1.5, 0%DIO, a), PDTBDT-TIPS-DTFBT-OD/PC61BM (1:1.5, 0%DIO, b) and PDTBDT-TIPS-DTNT-OD/PC71BM (1:1.5, 3%DIO, c).
